# How Can I Explain This to You? An Empirical Study of Deep Neural Network Explanation Methods

**Jeya Vikranth Jeyakumar, Joseph Noor, Yu-Hsi Cheng, Luis Garcia, Mani Srivastava**
`{vikranth94, jnoor, ellieyhc45, garcialuis, mbs}@ucla.edu`
University of California, Los Angeles
Los Angeles, California, USA

## Abstract

Explaining the inner workings of deep neural network models have received considerable attention in recent years. Researchers have attempted to provide human parseable explanations justifying why a model performed a specific classification. Although many of these toolkits are available for use, it is unclear which style of explanation is preferred by end-users, thereby demanding investigation. We performed a cross-analysis Amazon Mechanical Turk study comparing the popular state-of-the-art explanation methods to empirically determine which are better in explaining model decisions. The participants were asked to compare explanation methods across applications spanning image, text, audio, and sensory domains. Among the surveyed methods, explanation-by-example was preferred in all domains except text sentiment classification, where LIME's method of annotating input text was preferred. We highlight qualitative aspects of employing the studied explainability methods and conclude with implications for researchers and engineers that seek to incorporate explanations into user-facing deployments.

## 1  Introduction

In recent years, the explainability of deep neural network (DNN) models has come under scrutiny due to their black box nature [7, 14]. While it can approximate complex and arbitrary functions, studying its structure often provides little to no insight on the actual underlying mechanics. It is hard to look "into" the network and ascertain why specific features are selected over others during training. Although the impressive state-of-the-art performance cannot be disputed, it is increasingly insufficient to place confidence in inferences without justification. In particular, when considering deployments where models generate high-stakes predictions, sensitive decisions often mandate a sufficient accompanying explanation. For example, "Robot Radiologists" now provide superior MRI and X-Ray image classification in comparison to the average trained human expert [31]. A life-or-death diagnosis undeniably justifies the use of the best-performing model; however, it is unreasonable for either a patient or medical professional to simply accept an automated prediction at face value. With the inclusion of privacy regulations, including the GDPR "right to explanation," such an explanation is not only desirable, but also legally mandated [1].

While multiple definitions exist across the literature, we define an explanation as an image, text, or other visual aid that accompanies a prediction to offer intuition into the underlying reasons for the model output. Previous works have introduced several alternative techniques to provide such insight into a DNN model inference. Approaches span contrasting styles that focus on different model elements, e.g., the training dataset or the learned feature representations. Model-transparent approaches such as Grad-CAM++ [10] and saliency maps [36] highlight which particular input features triggered key activations within a model's weights. Model-agnostic methods such as LIME [32], SHAP [27], and Anchor [33] treat the model as a black-box and attempt to approximate the

relationship between the input sample and the output prediction. Finally, example-based methods [11, 21, 22, 24] offer instances from the training dataset in an attempt to capture the relationship between a given test input and the underlying training data that contributed to the model's decision.

Given the diverse suite of available methods, a challenge arises in determining which explanation is best suited to a particular application domain. The notion of a satisfying explanation is dictated primarily by the target audience. In the case of a radiology diagnosis, this includes both the patient and medical professional. Despite each method offering distinct benefits to a model developer, the non-technical end-user can easily become overwhelmed; a succinct yet clear explanation is needed. Selecting the most appropriate method for a deployment is currently limited by a lack of comprehensive empirical information elucidating the preferred explanation style according to the average end-user (i.e. individuals without machine learning expertise). Although there have been several works in the literature comparing and surveying different explanation methods [8, 9, 18, 20, 26, 40], to the best of our knowledge we are the first to explore the preferences of the general population toward explanation methods across multiple input domains. This insight can additionally help steer the efforts of researchers in designing new methods and improving upon existing solutions.

**Study Details.** We performed a comprehensive Mechanical Turk study comparing six of the most popular explanation methods across four distinct application domains to determine which styles are most preferred in understanding DNN model decisions. The explanations used include LIME, Grad-CAM++, Anchor, SHAP, saliency maps, and explanation-by-example. Applications incorporated data inputs including image, audio, text, and sensory analysis. Each study question compared two explanation methods for a test input, and the participant was instructed to select the method that they considered to offer a better explanation, thus providing a relative ranking. Responses were filtered out if the questionnaire was completed too quickly or if a sufficient fraction (20%) of the validation questions were incorrect. Every method was fine-tuned with optimized hyperparameters such that each was portrayed in the best manner possible. As there were no explanation-by-example implementation readily available, we developed and offer an open-source library *ExMatchina* for use.

**Study Results.** 4970 validated responses were collected from 455 participants. For the image, audio, and sensor application domains, the *explanation-by-example* style was the preferred method in **89.6%**, **70.9%**, and **84.8%** of the responses, respectively, when it was an available option, surpassing other methods by a statistically significant margin when considering bootstrap confidence. In the text application domain, *LIME*'s highlighted text with annotated sentiment prediction was the preferred explanation method in **70.4%** of the responses when it was an available option.

**Key Contributions.** This paper's contributions can be summarized as follows. First, we provide a unification, comparison, and analysis of existing explainability approaches for DNNs across various applications and input domains. Second, we present and discuss the results of a Mechanical Turk study[1] identifying the relative preference of explanation styles by an average non-technical end-user. Finally, we offer an open-source library *ExMatchina*[2], providing a readily available and widely applicable implementation of explanation-by-example.

## 2   Unifying Visual Explanation Methods Across Input Domains

Traditionally, DNN explanation frameworks emphasize their application over input domains that are naturally visualizable and understandable, e.g., image or text. However, DNNs are regularly applied across a wide class of domains, including time-series data (e.g., sensory data) that humans may struggle to reason about. Although most of the popular explanation methods are designed for images and text, prior works have shown that they can be successfully applied to time-series data [4, 17, 34].

Thus, we seek a unified representation such that an arbitrary input domain's explanation can be visualized and presented to end-users. In particular, we focus on the following representative dataset domains: image, text, audio, and sensory (ECG) data. These selected input domains are intended to provide insight into a subset of the popular DNN use cases. The unified representation supplies a common substructure to facilitate comparisons across multiple domains and enables a better understanding of the benefits of a particular approach.

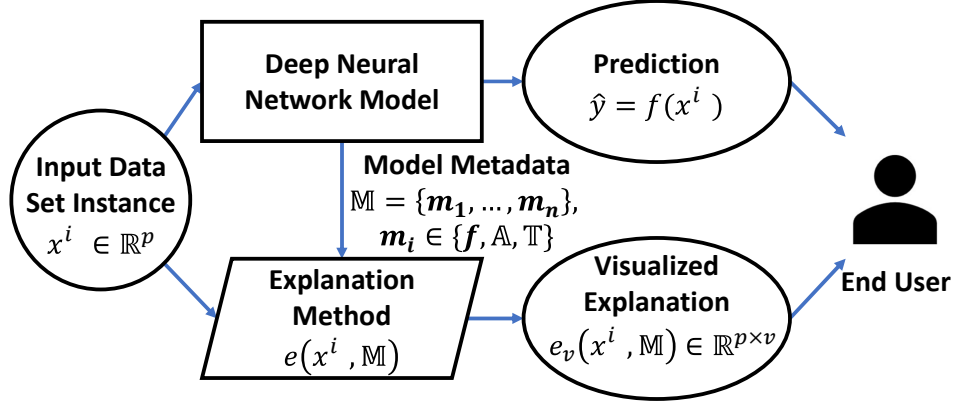

Figure 1: A unified representation of how we are drawing baseline comparisons for each visual explanation method, adapted from [8].

## 2.1 A Unified Representation of Visual Explanation Frameworks

Figure 1 summarizes our approach towards unifying visual explanation methods across input domains for a given pre-trained model and a test input. We focus on visual explanations as they are the predominant means for conveying explanations, e.g., via text or image representations. Our unification does not encompass adaptive NNs and methods exploring the intralayer and interlayer statistical properties [5, 12, 38].

A pre-trained DNN model $F$ maps a set of input data $\mathbb{X}$ for a domain $D$ to a set of labels $\mathbb{Y}$, i.e., $F : \mathbb{X} \to \mathbb{Y}$, where $\mathbb{X}_D = \{x_D^1, \ldots, x_D^n\}$ and the set $\mathbb{X}$ lies[3] in the domain space $\mathbb{R}^p$, i.e., $\mathbb{X} \in \mathbb{R}^p$. An explanation method requires a model metadata set $\mathbb{M} = \{m_1, \ldots, m_n\}$, where $m_i$ represents a component of the DNN model that is utilized. Existing frameworks typically employ the set of activations $\mathbb{A}$, the training dataset $\mathbb{T}$, or the black-box representation of model $f$. Certain explanations may only require the black-box representation of the model or have access to the activations and training dataset as well, i.e. $\mathbb{M} \subseteq \{f, \mathbb{A}, \mathbb{T}\}$.

For a given input data instance $x^i$ and a prediction $\hat{y} = F(x^i)$, an explanation method will generate an explanation $e(x^i, \mathbb{M})$. Traditionally, such an explanation would be in the form of a superimposed image on top of the original input data instance to highlight what features of the input "explain" the decision, i.e., both the explanation and the input were in the visual domain $V$ and the space $\mathbb{R}^v$ such that $x_V^i \in \mathbb{R}^v$ and $e(x^i, \mathbb{M}) \in \mathbb{R}^v$. For any given task where the input domain is in a non-visual space, $\mathbb{X} \in \mathbb{R}^p \setminus \mathbb{R}^v$, the input and its explanation should be mapped to the visual space such that

$$e_v(x^i, \mathbb{M}) \in \mathbb{R}^{p \times v}. \tag{1}$$

## 2.2 Superimposition Based Explanation Methods

A vast majority of explanation methods focus on visualizing an explanation by superimposing values onto the original test input data instance. While most of these frameworks already support the image and text domains, we also sought to provide visualized explanations for tasks in the audio and sensory domains. Generally, sensory and audio data are time-series that can be plotted for visualization. While DNN models and associated explanations are trained and inferenced on the raw time-series data, explanations are ultimately presented in visual form. That is, explanation outputs are inherently mapped to images. The value at each time step of a time-series data is analogous to pixels in the image and, therefore, the explanation methods can be applied by highlighting time instances or slices on the plots of the signal waveforms. Thus, we will describe how this intuition can lead to visualizations in multiple domains for both *model-agnostic* and *model-transparent* explanation methods. Generally, both categories will result in a superimposition of an explanation $e(x^i, \mathbb{M})$ onto an input instance $x^i$, i.e.,

$$e_v(x^i, \mathbb{M}) = g(x^i) + g(e(x^i, \mathbb{M})) \tag{2}$$

| Explanation Style | Explanation Method | Domains | | | |
|---|---|---|---|---|---|
| | | **Image** | **Text** | **Audio** | **Sensory data** |
| Superimposition over test input | LIME | ✓ | ✓ | ✗ | ✗ |
| | Anchor | ✓ | ✓ | ✗ | ✗ |
| | SHAP | ✓ | ✓ | ◯ | ◯ |
| | Saliency Maps | ✓ | ✗ | ◯ | ◯ |
| | Grad-CAM++ | ✓ | ✗ | ◯ | ◯ |
| Explanation-by-Example | ExMatchina | ✓ | ✓ | ✓ | ✓ |

Table 1: Comparing explanation methods across different input domains. Checkmarks (✓) indicate that a method is explicitly designed for a domain. Circles (◯) indicate that a method was able to be successfully adapted to a domain. Crossmarks (✗) indicate that adapting a method to the specified domain is non-trivial; we chose to exclude these evaluations to avoid potentially inaccurate representations that portray these methods in a suboptimal light.

where $g$ is a function that projects both the input space as well as the original explanation onto the visual space, e.g., a plot of an ECG input sample.

**Model-agnostic methods.** Model-agnostic explanation methods treat the model as a black-box, i.e., $\mathbb{M} = \{f\}$. The general approach is to approximate the relationship between the input and the output prediction. Surrogate methods such as LIME [32] create a proxy model that is inherently interpretable to produce a local approximation of the relationship between a prediction and perturbed instances of the input data. Anchor [33] uses a similar perturbation-based approach to approximate the relationship between the model's prediction and the input data via *high-precision if-then rules*. Likewise, SHAP [27] computes the contribution of each feature of an input test instance to the output prediction. In particular, SHAP uses a game-theory based approach by computing Shapley values for each feature.

**Model-transparent methods.** Model-transparent superimposition methods have full access to the DNN pipeline and typically focus on the relationship between an input instance, an output instance, and the associated activations of the hidden layers, i.e., $\mathbb{M} = \{f, \mathbb{A}\}$. These methods similarly aim to highlight the features of the input that are important to the output classification. Popular methods such as saliency maps [36] visualize the gradient of the output classification with respect to the pixels of the input image. Grad-CAM++ [10] superimposes a heatmap on the regions of important input features computed by the weighted gradients of an output classification with respect to the final convolutional layer of a CNN model.

For both model-agnostic and model-transparent methods, the raw data is provided to the DNN model and the associated explanation method without modifications[4]. Upon generating the explanation, the raw data will be visualized and the explanation is superimposed on the visualized representation.

## 2.3 Training Data Based Explanation Methods

In contrast to methods that project explanations onto the input space, techniques such as explanation-by-example fall under a category that project explanations across the underlying training data or other representative prototype examples. There is a wide variety of explanation by example frameworks that may focus on various aspects of the DNN model as well as the training dataset, i.e., $\mathbb{M} = \{f, \mathbb{A}, \mathbb{T}\}$. Protoype methods [29] and their associated critics [22] focus on the distributions of the input dataset with respect to an inference to generate prototype examples for a particular input. Techniques such as influence functions [24] incorporate the relationship between the training dataset and the model weights at training time to identify the dataset samples that are most *influential* for a particular classification.

As explanation-by-example frameworks generate a set of examples as an explanation, the mapping of an explanation of any domain to the visual domain is more straightforward. Generated examples are visualized in the same manner by which an input data instance is visualized. Thus, if the general function for explanation by example frameworks is defined as

$$e(x^i, \mathbb{M}) = \mathbb{E} = examples(x^i), \tag{3}$$

then the associated visualized explanation for all domains will be

$$e_v(x^i, \mathbb{M}) = g(E), \forall E \in \mathbb{E}. \tag{4}$$

To the best of our knowledge, there is currently no openly available implementation of explanation-by-example. However, as this style of explanation has received considerable attention in recent work [2, 22, 24, 29], we believed it important to include in the study. Our open-source implementation of explanation-by-example, *ExMatchina*, provides the nearest matching data samples from the training dataset as representative examples. Nearest examples were selected by comparing feature activations at the last convolutional layer.

The nearest matching examples from the training set $\mathbb{T}$ is defined as the training data that has the highest cosine similarity with the test input $x$ in their activations:

$$examples(x) = \max_{t \in \mathbb{T}} \cos(A^k(x), A^k(t)) = \max_{t \in \mathbb{T}} \frac{\sum_{i=1}^n A_i^k(x) A_i^k(t)}{\sqrt{\sum_{i=1}^n (A_i^k(x))^2} \sqrt{\sum_{i=1}^n (A_i^k(t))^2}} \tag{5}$$

where $A^k$ represents the visualization of the $k^{th}$ feature map as a vector of activation values. The use of cosine similarity as a distance metric for selecting the nearest neighbors is validated by Papernot and McDaniel [29]. For multiple examples, the $\max_N$ samples can be obtained.

We summarize the domains currently supported by state-of-the-art frameworks in Table 1.

## 3 Study Methodology

We conducted four separate Amazon Turk studies, one for each dataset. Study questions were formed in the following manner: first, a random test input and model-predicted class were presented along with two randomly selected explanations. Participants were asked to select which of the two available methods offered a better explanation for the provided model prediction. Due to the variability in the amount of time required to parse a test input and the associated explanations, the number of questions each participant answered varied across each study; the image classification questionnaire consisted of 15 comparisons, 12 for text classification, 6 for keyword classification, and 12 for the ECG-based arrhythmia heartbeat classification task.

**Validating Responses.** Two filtering criteria were included to eliminate participants providing illegitimate responses. First, participants that provided responses faster than a minimum threshold required to quickly read the survey were removed to exclude submissions auto-completed by bots. Second, each test input was accompanied by a validation question asking if they agree with the model prediction without providing the true label to cross-reference whether a participant was willing and able to comprehend the test input provided; those that failed 20% of these validation questions were eliminated from the published results. As the average participant is unlikely to have sufficient insight into the ECG sensor inputs, validation questions in that particular questionnaire offered the true label and simply asked participants to select whether or not the model was correct in its prediction.

**IRB Exemption and Compensation.** This research study has been certified as exempt from review by the IRB per 45 CFR 46.101, category 2 (UCLA IRB#20-000893). Participants were compensated at a rate of 15 USD per hour.

### 3.1 Tasks and Datasets

In an attempt to capture a wide array of common DNN use cases, we selected well-known classification datasets across each of the surveyed input domains, as depicted in Table 2. The test samples included in the study questionnaire were randomly selected from each domain's test set. The Cifar10

| Task | Image Recognition | Sentiment Analysis | Key Word Detection | Heartbeat Classification |
|---|---|---|---|---|
| **Domain** | Image | Text | Audio | Sensory data (ECG) |
| **Dataset** | Cifar-10 | Sentiment140 | Speech Commands | MIT-BIH Arrhythmia |
| **Classes** | 10 | 2 | 10 | 5 |

Table 2: An overview of the application tasks and datasets used in our study

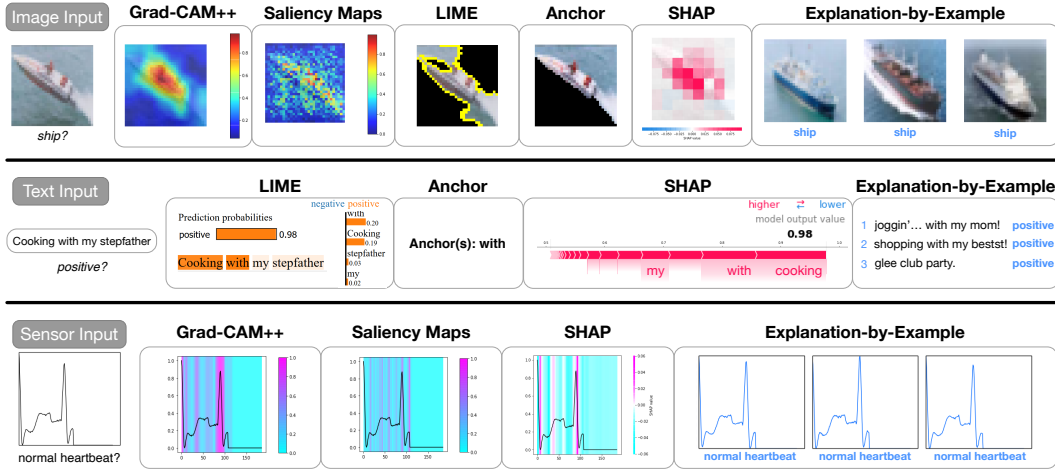

Figure 2: Depiction of surveyed explanation methods for image, text, and ECG input.

dataset [25] was selected for image data, the Sentiment140 dataset [19] for text, Google's Speech Commands dataset [39] for audio, and the the MIT-BIH Arrhythmia ECG Dataset [28] for sensory data. Cifar10 is a classic image classification dataset with 10 different classes. The text domain application performed sentiment analysis over Sentiment140, consisting of tweets with associated positive or negative sentiment; the shorter text length was explicitly desired to maximize participant engagement. For audio keyword recognition, the ten most common keywords of one-second long utterances were extracted from Google's speech commands dataset. In the sensory data domain, ECG data represents a class of sensory values that are relatively familiar and recognizable to the average individual. A heartbeat classification application was derived from the MIT-BIH Arrhythmia Dataset. Tabular data were excluded from our study as models are typically derived using interpretable linear models or decision trees instead of DNNs.

## 3.2 Models and Explanations

A CNN (Convolutional Network Network) model was trained for each of the four tasks. Batch-normalization and dropout layers were used in all models. The trained models achieved f-1 scores of 87.8%, 83.5%, 90.2%, and 86.2% on Image classification, Text Sentiment Analysis, Key Word detection, and ECG Heartbeat classification, respectively. Screenshots of the explanations for image, text, and sensor data are presented in Figure 2. More specific details on the model architecture, hyper-parameters, and audio data explanations are available in the supplementary material.

### 3.2.1 Configuring and Optimizing Explanation Methods

To provide a comparative analysis across various approaches to visual explanations, we selected a subset of methods that were most commonly explored in related works and explainability studies [8, 9, 18]: Grad-CAM++, saliency maps, LIME, Anchor, SHAP, and *ExMatchina*. Though several heatmap methods exist [16, 30, 35], we selected Grad-CAM++ as the representative as it provides a more localized and compact explanation when compared to Grad-CAM, from which it is based on [10]. Our open-source implementation of explanation-by-example, *ExMatchina*, uses NN-query based on a number of prior works that advocate for a similar approach [17, 29].

Model transparent explanation methods including Grad-CAM++, saliency maps, and *ExMatchina* require the specification of which model layer to use in generating explanations. To ensure a fair comparison for these methods, we universally selected the activations after the last convolutional layer to uniformly capture the information extracted by the convolutional layers. This approach is validated by previous work [10] and our empirical observation that the last convolutional layers generally provided the best explanations.

LIME and Anchor [23] required various hyperparameter tuning to generate acceptable explanations on images. The Felzenszwalb segmentation algorithm [15] was used to segment images into super-pixels.

SHAP has several different approximation methods to choose from. DeepExplainer was used on text, and GradientExplainer was used on image, audio, and ECG, as we empirically observed these to provide the best explanations for the datasets in our study.

Not all of the methods studied were intended for the suite of explored datasets. In particular, Grad-CAM++ and saliency maps are intended for image classification only. However, by treating the ECG and audio test samples as a one-dimensional image with only a single channel, normalized between 0 and 1, we were able to extract explanations for these datasets using these methods. While we were able to successfully adapt the Grad-CAM++, saliency map, and SHAP methods for audio and sensory domains, it was non-trivial to adapt the saliency map and Grad-CAM++ to text and LIME and anchor to audio and ECG. To avoid potentially inaccurate representations that risk portraying methods in a suboptimal manner, we opted to exclude their application to these respective domains for this study.

Explanation toolkits that provided a built-in display method were used as is, and visual depictions were generated for those that did not provide such a solution. For example, as SHAP was used on ECG and audio data, which cannot be illustrated in the exact same manner as an image, we created plots and highlighted the slices with the SHAP value gradients to illustrate the time slices that contributed positively or negatively. Similarly, in applying Grad-CAM++ and saliency maps to ECG and audio data, we highlighted the background of the time slices with the associated heatmap. In order to make audio data more comprehensible for the survey participants, we generated videos for each test sample and explanation. The waveform of the sample was plotted and overlaid with the audio track. When played, the video displayed a vertical bar that translates horizontally to accordingly indicate which part of the waveform is significant based on the particular explanation.

## 4 Study Results & Discussion

Table 3 presents the aggregated results of the Mechanical Turk study. After questionnaire validation, 4970 responses were collected across 455 individuals. Explanation-by-example was the largely preferred explanation style for image, audio, and sensory data classification; when *ExMatchina* was an available explanation, participants selected it 89.6%, 70.9%, and 84.8% of the time, respectively. In the text input domain, LIME was the preferred explanation method, as it was preferred 70.4% of the time. The presented confidence intervals are calculated using the bootstrap method as described in [13] with 95% confidence interval. These results should not be interpreted to conclude that any of these methods are inherently superior, but instead indicate that these are the methods most appealing to those who may not possess knowledge of machine learning, i.e. the "non-expert" layperson.

### 4.1 Usability and Stability of Explanations

LIME and Anchor were particularly unstable due to the inconsistent results generated across multiple runs. An additional contributor to their poor performance in the image domain was the reliance on segmentation algorithms to first split the image into superpixels. Selecting across the multiple available algorithms (e.g., Felzenszwalb, Slic, Quickshift, and Compact watershed) increased the overhead of ensuring the overall method was applied effectively. In contrast, segmentation was not necessary in the text domain as the granularity of words provide a natural unit of semantic value. Nevertheless, LIME and Anchor's explanations were still unstable across repeated runs; different explanations can be generated each time these methods are run on the same test sample. Moreover, each application required experimentation with different kernel and hyper-parameters settings to optimize against the generated explanations. In summary, these highly tunable methods incur increased configuration complexity and a higher probability of suboptimal specification.

| Explanation Method | Image Study | Text Study | Audio Study | Sensor Study |
|:---:|:---:|:---:|:---:|:---:|
| LIME | 47.7 ± 4.5% | **70.4 ± 3.6%** | - | - |
| Anchor | 38.9 ± 4.3% | 25.8 ± 3.5% | - | - |
| SHAP | 33.7 ± 4.3% | 59.9 ± 3.8% | 34.7 ± 4.8% | 32.8 ± 3.3% |
| Saliency Maps | 39.4 ± 4.3% | - | 46.1 ± 5.1% | 40.4 ± 3.5% |
| Grad-CAM++ | 50.8 ± 4.5% | - | 48.1 ± 5.3% | 42.0 ± 3.5% |
| ExMatchina | **89.6 ± 2.6%** | 43.7 ± 3.9% | **70.9 ± 4.7%** | **84.8 ± 2.5%** |

Table 3: Results of the Mechanical Turk study evaluating user preference for DNN explanation methods across image, text, audio, and sensory input domains. Survey questions individually compare two methods at a time, with each explanation compared to all other available methods equally. Results indicate the rate by which users selected a particular method when it is an available explanation, with 95% bootstrap confidence intervals.

In contrast, the other surveyed methods including Grad-CAM++, saliency maps, SHAP, and explanation-by-example did not require such fine-grained hyperparameters tuning, making them significantly easier to use. The only configuration parameter was the model layer to use in generating explanations. Moreover, these methods provide stable explanations; repeated runs over the same test instance lead to the same result.

## 4.2 Idealized vs Actualized Explanations

An individual's expectation of an explanation is sometimes different than those provided by the studied methods. In particular, superimposition explanations such as Grad-CAM++, saliency maps, and SHAP highlight input features that are most important to the predicted classification. However, these features may be in stark contrast to human intuition; similarly, Anchor and LIME would occasionally mask features that humans think are explicitly important. For example, given an image of an airplane in the sky, superimposition methods might highlight the sky to indicate that the sky is the most important detail determining that the object is an airplane; however, a human might prefer that the airplane is the defining characteristic leading to the airplane labeling. Similarly, when explaining audio data these methods would occasionally emphasize sections of the input that contained no speech (i.e., noise, background audio, or silence). In ECG data, flat sections may be highlighted instead of the actual heartbeat spikes. An ironic downside of using superimposition methods is that they show features which the models think are important, but which humans might not focus on, ultimately leading to the explanation method being generally less preferable.

Similarly, explanation-by-example should ideally produce examples that are always highly similar to the test input. In the image, audio, and ECG domains, we found that the nearest training examples repeatedly offered an intuitive and semantically similar mapping of training data, associated labels, and model inference. For example, nearest image examples commonly depicted the predicted class with a visually similar position, orientation, and relative sizing. Cases where the training examples could be considered dissimilar (i.e. "bad" explanations) were relatively rare. This consistency of explanation is a likely contributor to the preference of this method over the alternatives. In the audio and ECG domains, training examples were often impressively similar, including matching signal noise, event length, and position within the sample window. However, the semantic similarity of explanation-by-example is fundamentally limited by the quality of the training data; a lack of similar examples would necessarily lead to a subpar explanation [22]. This was most obviously apparent in the text domain, where the nearest training examples to a test input may be seemingly unrelated.

In the text domain, LIME in particular offers an intuitive annotation of the text with associated sentiments and expected probabilities. The accuracy by which the annotated sentiments consistently matched human intuition, combined with its assembly into a natural visual interface, are likely key contributors to its success in explaining text data. Interestingly, LIME and Anchor occasionally assign sentiment to words that are neutral according to human intuition; for example, assigning positive or negative sentiment to words such as "at" or "with".

### 4.3 Privacy Risks

Unlike superimposition methods, explanation-by-example mandates access to a set of reference data (e.g., training data) by which to generate nearest examples. This poses privacy risks regarding potentially exposing personally identifiable information, particularly when operating over sensitive training data. Those seeking to employ explanation-by-example methods in deployments should ensure that the examples offered do not violate user privacy. In contrast, model-transparent methods that reveal model weights offer a reduced attack surface for inferring underlying data; model-agnostic methods that rely only on a test input and the model prediction function are able to mostly circumvent these concerns [37]. In certain instances, differential privacy techniques can be used to anonymize training data when revealing training examples [3, 6]. While we note that this may impact the quality of explanation, studying its quantitative effect is left to future work.

## 5    Conclusion

Successfully explaining deep neural network models hinges upon having an effective means of communicating the inner-workings of these complex processes. In certain domains, the explicit need for interpretable models outweighs the performance gains of black-box neural network methods. However, in the cases where the performance offered by DNNs are needed, it is important that they are accompanied by explanations that provide satisfying insight into model behavior.

Our study across hundreds of participants conclude that explanation-by-examples and LIME are the currently preferred explanation styles according to the average non-technical end-user. In input domains spanning visual, audio, and sensory data, explanation by nearest training examples offer users an opportunity to compare features across a test input and similarly mapped ground-truth examples. In the text domain, LIME's method of decomposing and annotating test inputs provides an intuitive visual approach to text classification. Although the other studied methods can be retargeted across many of the surveyed input domains, they failed to provide a more desirable explanation. Future efforts in designing DNN explanations are certain to challenge the current baseline; nevertheless, we hope our results and discussion bring insights empowering researchers and engineers to incorporate effective means of communicating complex inferences with the end-user.

## Broader Impact

We would like to begin by praising the community for it's recent shift of attention to the study and design of high-quality model explanations, particularly when attempting to provide insight into inferences that cannot be easily interpreted. This essential direction of research indicates that scientists and technologists care about making complex methods more accessible and understandable. Encouraging a world where the advanced computational techniques used to influence modern society are generally understandable empowers us all to steer its direction with clearer vision toward a more desirable state. At the very least, we can be more prepared to avoid negative outcomes.

**Fear of the Unknown.** As deep neural networks have brought performance improvements that were seemingly unimaginable, the general public remains largely ignorant of what makes these approaches so powerful; to them, it's equivalent to magic. This is popularly conveyed in sensational headlines and the fictional portrayals of dystopian futures, laying somewhere at the intersection of *Black Mirror*, Orwell's *1984*, and *Terminator*'s Skynet. Nevertheless, these fears underlay legitimate concerns that motivate privacy preserving regulations including GDPR and CCPA, not to mention the inclusion of the NeurIPS Broader Impact Statement. The continuing efforts to research effective methods of ensuring explainability of these models can help partially alleviate this tension, particular if we succeed in providing an honest (yet comprehensible) representation of the underlying method. This is the motivation for our work.

**Positive Impact.** The stated objective and ideal impact of this paper is relatively straightforward; we intend to empower the community with knowledge of the average end-user perspective. In this way, we might all have a better understanding of the right approach to increase transparency and effectively communicate with the public, thereby offering a bridge between the technologist and non-technical layperson.

**Unintended Risks.** One of our research conclusions is that exposing subsets of the underlying training data is an effective means of justifying complex model predictions. This poses inherent privacy risks. A necessary complement to explanation-by-example are techniques to anonymize and sanitize personally identifiable information from the revealed training data. Thankfully, the active body of literature surrounding differential privacy offers techniques that can be employed when needed to ensure these privacy violations do not occur. It is critical that the noble effort to explain complex models do not unintentionally harm the individuals potentially comprising the training data.

One possible side-effect of this paper is that researchers may be discouraged from advancing explainability if faced with a published relative ranking. On the very contrary, we instead hope it inspires novel solutions and brings increased attention to the importance of this problem. This paper is by no means meant to provide closure, but instead serve as a stepping stone detailing the current landscape for community reflection and reevaluation.

**Moving Forward.** The only way that we can ensure that our community does not bring harm to the public is through transparency and honesty. Of course, this is impossible without an effective means of mutual communication — one that can be understood by all. This paper is an attempt to elucidate the methods and styles of presentation that offer this universal language by which we can create a more informed society.

## Acknowledgments and Disclosure of Funding

The research presented in this paper is supported in part by the CONIX Research Center, one of six centers in JUMP, a Semiconductor Research Corporation (SRC) program sponsored by DARPA, the U.S. Army Research Laboratory and the U.K. Ministry of Defence under Agreement Number W911NF-16-3-0001 and the National Science Foundation (NSF) under award # CNS-1822935, and by the National Institutes of Health (NIH) award # P41EB028242 for the mDOT Center. The views and conclusions contained in this document are those of the authors and should not be interpreted as representing the official policies, either expressed or implied, of the NSF, the NIH, the U.S. Army Research Laboratory, the U.S. Government, the U.K. Ministry of Defence or the U.K. Government. The U.S. and U.K. Governments are authorized to reproduce and distribute reprints for Government purposes notwithstanding any copyright notation hereon.

## Footnotes

[1]https://github.com/nesl/Explainability-Study

[2]https://github.com/nesl/ExMatchina

[3]For the sake of clarity, we omit the notation $D$ as we assume the input from all domains has been translated to the real space, e.g., a text input is first converted to an integer representation.

[4]Although an approach like LIME may require segmentation of the data, we assume these procedures to be subsumed by the function $e(x^i, \mathbb{M})$.

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
