[Supplementary Material 1 · text.pdf]



# Thanks for participating in our study!

You were selected as a possible participant in this study because you are an MTurk worker, and your participation in this research study is voluntary.

**Please read:**

For each task, you'll see a **tweet** and an **Artificial Intelligence (AI) prediction** of whether the tweet was positive or negative.

First, we want to know if you think the AI was correct in its decision.

Next, there will be two methods explaining why the AI made this decision.

We want your opinion on **which is a better explanation** for this decision.

Please be sure to select one of the methods for every image.

Note: Sometimes it may be unclear if the tweet is actually positive or negative. Please select whichever makes more sense to you.

---

## Task 1

**An AI algorithm thinks this is a positive tweet.**

**Correctness:** Do you think the AI classified this tweet correctly?

○ Yes

○ No

### Explanation 1: Keywords

The anchor words are the specific words that the AI thinks are most important to the tweet being positive.

### Explanation 2: Highlighted Tweet

The AI highlighted the parts of the tweet that are most important.

**Which explanation is better?**

◯ Keywords (top)

◯ Highlighted Tweet (bottom)

# Task 2

**An AI algorithm thinks this is a negative tweet.**

Power is out only at my house

**Correctness:** Do you think the AI classified this tweet correctly?

◯ Yes

◯ No

**Explanation 1: Training Examples**

The AI thinks this tweet is very similar to other tweets that it was told are negative.

SIMILAR TWEET #1 (negative):
Tonights truffle project did not go as planned

SIMILAR TWEET #2 (negative):
My smelling salts exploded in my purse

SIMILAR TWEET #3 (negative):
My Dad's brother died.  Come cuddle me.

**Explanation 2: Keywords**

The anchor words are the specific words that the AI thinks are most important to the tweet being negative.

Anchor(s): only

**Which explanation is better?**

◯ Training Examples (top)

◯ Keywords (bottom)

# Task 3

**An AI algorithm thinks this is a negative tweet.**

Noon+ wake up streak comes to an end tomorrow

**Correctness:** Do you think the AI classified this tweet correctly?

○ Yes

○ No

## Explanation 1: Training Examples

The AI thinks this tweet is very similar to other tweets that it was told are negative.

SIMILAR TWEET #1 (negative):
Sign up for Internet World is NOT user friendly

SIMILAR TWEET #2 (negative):
Its evening here in India! Right now, Studying for exams!

SIMILAR TWEET #3 (negative):
Calling it a night. Waking up very early tomorrow

## Explanation 2: Highlighted Tweet

The AI highlighted the parts of the tweet that are most important.

Prediction probabilities

negative    0.81
positive    0.19

negative        positive

end      0.39
an       0.14
to       0.10
tomorrow 0.07
Noon     0.06
up       0.03

**Text with highlighted words**

Noon+ wake up streak comes to an end tomorrow

**Which explanation is better?**

○ Training Examples (top)

○ Highlighted Tweet (bottom)

# Task 4

**An AI algorithm thinks this is a negative tweet.**

Happy Father's Day to all of the fathers out there. Wishing  wasn't out of town today.

**Correctness:** Do you think the AI classified this tweet correctly?

○ Yes

○ No

### Explanation 1: Training Examples

The AI thinks this tweet is very similar to other tweets that it was told are negative.

SIMILAR TWEET #1 (negative):
Happy Father's Day to all of the deserving Fathers out there!  I miss my Daddy today...

SIMILAR TWEET #2 (negative):
Trying to remove a tick from my leg. Shoulda checked myself last night

SIMILAR TWEET #3 (negative):
Well you didn't come with the VGA Adapter like the Auction stated, so no 480P for me or you

### Explanation 2: Positive/Negative Analysis

The AI decision thought the red words were positive and the blue words were negative.

**Which explanation is better?**

○ Training Examples (top)

○ Positive/Negative Analysis (bottom)

## Task 5

**An AI algorithm thinks this is a negative tweet.**

i'm going to meet my Director's Guild Family tomorrow, I miss them so much!

**Correctness:** Do you think the AI classified this tweet correctly?

○ Yes

○ No

## Explanation 1: Highlighted Tweet

The AI highlighted the parts of the tweet that are most important.

Prediction probabilities

negative    0.90

positive    0.10

negative    positive

miss    0.68

meet    0.09

m    0.07

going    0.06

Guild    0.05

much    0.04

**Text with highlighted words**

i'm going to meet my Director's Guild Family tomorrow, I miss them so much!

## Explanation 2: Positive/Negative Analysis

The AI decision thought the red words were positive and the blue words were negative.

higher ⇄ lower

model output value

**0.10**

base value

0.0    0.1    0.2    0.3    0.4    0.5    0.6

going    meet    miss    i    so    much    them

**Which explanation is better?**

○ Highlighted Tweet (top)

○ Positive/Negative Analysis (bottom)

# Task 6

**An AI algorithm thinks this is a positive tweet.**

but of course  good things come in small packages woof 2 u & Quinn

**Correctness:** Do you think the AI classified this tweet correctly?

○ Yes

○ No

**Explanation 1: Keywords**

The anchor words are the specific words that the AI thinks are most important to the tweet being positive.

Anchor(s): course

**Explanation 2: Positive/Negative Analysis**

The AI decision thought the red words were positive and the blue words were negative.

**Which explanation is better?**

○ Keywords (top)

○ Positive/Negative Analysis (bottom)

# Task 7

**An AI algorithm thinks this is a positive tweet.**

actually we are out  but im eatin leftovers from last nights PopEyes! its yummy

**Correctness:** Do you think the AI classified this tweet correctly?

○ Yes

○ No

**Explanation 1: Keywords**

The anchor words are the specific words that the AI thinks are most important to the tweet being positive.

Anchor(s): **yummy**

## Explanation 2: Highlighted Tweet

The AI highlighted the parts of the tweet that are most important.

Prediction probabilities

negative    0.13
positive    0.87

negative          positive

yummy 0.21
but 0.12
last 0.08
nights 0.07
eatin 0.06
actually 0.06

**Text with highlighted words**

actually we are out  but im eatin leftovers from last nights PopEyes! its yummy

**Which explanation is better?**

◯ Keywords (top)

◯ Highlighted Tweet (bottom)

# Task 8

### An AI algorithm thinks this is a negative tweet.

Day 2 of #bbplan on hold. Business link were supposed to call me for a tel appointment this morning. No free time to take a call now

**Correctness:** Do you think the AI classified this tweet correctly?

◯ Yes

◯ No

## Explanation 1: Training Examples

The AI thinks this tweet is very similar to other tweets that it was told are negative.

SIMILAR TWEET #1 (negative):
please say a prayer for my hubby. Delta lost his 2 bags since Friday and we really can't
afford to buy all new clothes and shoes

SIMILAR TWEET #2 (negative):
my yard outside my window is flooded. I have my plants outside to conserve watering.
Can't open sliding door b/c of dog.

SIMILAR TWEET #3 (negative):
i wanna go get my tattoo now but i have no
$$$/dinero/cash/bucks/green/benjamins/change/other synonyms for money...

## Explanation 2: Keywords

The anchor words are the specific words that the AI thinks are most important to the tweet being negative.

Anchor(s): No

**Which explanation is better?**

◯ Training Examples (top)

◯ Keywords (bottom)

# Task 9

**An AI algorithm thinks this is a negative tweet.**

now I donno how to get there to see ya. Google can't figure out a route.  sadness

**Correctness:** Do you think the AI classified this tweet correctly?

◯ Yes

◯ No

## Explanation 1: Training Examples

The AI thinks this tweet is very similar to other tweets that it was told are negative.

SIMILAR TWEET #1 (negative):
not been well today so no work today or tomrrow, 2 days off but no enjoyment

SIMILAR TWEET #2 (negative):
sooo sleepy. dont want to be at work but i still have 2 hours left. tummyache

SIMILAR TWEET #3 (negative):
It wasn't rainy here in DE..it was just kinda cold and cloudy and depressing

## Explanation 2: Highlighted Tweet

The AI highlighted the parts of the tweet that are most important.

Prediction probabilities

negative    0.99

positive    0.01

negative           positive

sadness
0.18

donno
0.08

t
0.07

out
0.04

ya
0.03

route
0.02

**Text with highlighted words**

now I donno how to get there to see ya. Google can't figure out a route.  sadness

**Which explanation is better?**

◯ Training Examples (top)

◯ Highlighted Tweet (bottom)

---

## Task 10

**An AI algorithm thinks this is a negative tweet.**

One of my new fish is dieing!

**Correctness:** Do you think the AI classified this tweet correctly?

◯ Yes

◯ No

### Explanation 1: Training Examples

The AI thinks this tweet is very similar to other tweets that it was told are negative.

SIMILAR TWEET #1 (negative):
Going to Maryland! To a funeral...

SIMILAR TWEET #2 (negative):
The download page of beta is empty.

SIMILAR TWEET #3 (negative):
Undoing my perfect, intricate hair for prom is painful

### Explanation 2: Positive/Negative Analysis

The AI decision thought the red words were positive and the blue words were negative.

**Which explanation is better?**

○ Training Examples (top)

○ Positive/Negative Analysis (bottom)

# Task 11

**An AI algorithm thinks this is a positive tweet.**

> Please follow @Milano    i know its not #followfriday but do it anyway!

**Correctness:** Do you think the AI classified this tweet correctly?

○ Yes

○ No

## Explanation 1: Highlighted Tweet

The AI highlighted the parts of the tweet that are most important.

**Text with highlighted words**

Please follow _Milano   i know its not #followfriday but do it anyway!

## Explanation 2: Positive/Negative Analysis

The AI decision thought the red words were positive and the blue words were negative.

**Which explanation is better?**

○ Highlighted Tweet (top)

○ Positive/Negative Analysis (bottom)

# Task 12

**An AI algorithm thinks this is a negative tweet.**

> Hd enough of tit for tat with the kids. I'll nvr win. They came up with the silliest games like "who will blink first?" No bets on me

**Correctness:** Do you think the AI classified this tweet correctly?

○ Yes

○ No

## Explanation 1: Keywords

The anchor words are the specific words that the AI thinks are most important to the tweet being negative.

**Anchor(s): nvr, No**

## Explanation 2: Positive/Negative Analysis

The AI decision thought the red words were positive and the blue words were negative.

**Which explanation is better?**

◯ Keywords (top)

◯ Positive/Negative Analysis (bottom)

Note: only click submit once, or your entry may be lost.

Submit



[Supplementary Material 2 · ecg.pdf]



# Thanks for participating in our study!

You were selected as a possible participant in this study because you are an MTurk worker, and your participation in this research study is voluntary.

## Please read:

For each task, you'll see a **heartbeat** from a patient with Arrhythmia. An **Artificial Intelligence (AI) prediction** is trying to determine the type of heartbeat.

There are 5 types of heartbeats:

- **N (normal, non-ectopic):** a normal heartbeat
- **S (supraventricular ectopic):** a premature beat that occurs in the atrium
- **V (ventricular ectopic):** a premature beat that occurs in the ventricles
- **F (fusion):** a fusion of a normal (N) and ventricular (V) beat
- **Q (unknown):** a noisy beat with unknown classification

First, we want to know if the AI was correct in its decision.

Next, there will be two methods explaining why the AI made this decision.

We want your opinion on **which is a better explanation** for this decision.

Please be sure to select one of the methods for every task.

---

# Task 1

## An AI algorithm thinks this is a N beat.

N?

---

**Correctness:** The heartbeat is actually a N heartbeat. Did the AI classify this heartbeat correctly?

○ Yes

○ No

---

## Explanation 1: Heatmap

Here is a heatmap of where in the heartbeat the AI thinks are most important.

## Explanation 2: Positive/Negative Analysis

The AI decision was influenced positively by the magenta areas and negatively by blue areas.

> **Which explanation is better?**
>
> ○ Heatmap (top)
>
> ○ Positive/Negative Analysis (bottom)

---

# Task 2

## An AI algorithm thinks this is a N beat.

> **Correctness:** The heartbeat is actually a N heartbeat. Did the AI classify this heartbeat correctly?
>
> ○ Yes
>
> ○ No

## Explanation 1: Hot Pixels

Magenta indicates the exact moments the AI thinks are important.

## Explanation 2: Training Examples

The AI thinks this heatbeat is similar to other already known examples of N heartbeats.
They make look very similar but they are actually all different heartbeats.

**Which explanation is better?**

○ Hot Pixels (top)

○ Training Examples (bottom)

# Task 3

**An AI algorithm thinks this is a N beat.**

N?

**Correctness:** The heartbeat is actually a N heartbeat. Did the AI classify this heartbeat correctly?

◯ Yes

◯ No

## Explanation 1: Training Examples

The AI thinks this heatbeat is similar to other already known examples of N heartbeats.
They make look very similar but they are actually all different heartbeats.

## Explanation 2: Positive/Negative Analysis

The AI decision was influenced positively by the magenta areas and negatively by blue areas.

<div style="border:1px solid;padding:10px">

**Which explanation is better?**

○ Training Examples (top)

○ Positive/Negative Analysis (bottom)

</div>

# Task 4

**An AI algorithm thinks this is a S beat.**

<div style="border:1px solid;padding:10px">

**Correctness:** The heartbeat is actually a S heartbeat. Did the AI classify this heartbeat correctly?

○ Yes

○ No

</div>

## Explanation 1: Hot Pixels

Magenta indicates the exact moments the AI thinks are important.

## Explanation 2: Positive/Negative Analysis

The AI decision was influenced positively by the magenta areas and negatively by blue areas.

**Which explanation is better?**

○ Hot Pixels (top)

○ Positive/Negative Analysis (bottom)

---

# Task 5

## An AI algorithm thinks this is a N beat.

**Correctness:** The heartbeat is actually a N heartbeat. Did the AI classify this heartbeat correctly?

○ Yes

○ No

## Explanation 1: Hot Pixels

Magenta indicates the exact moments the AI thinks are important.

## Explanation 2: Heatmap

Here is a heatmap of where in the heartbeat the AI thinks are most important.

**Which explanation is better?**

○ Hot Pixels (top)

○ Heatmap (bottom)

---

# Task 6

## An AI algorithm thinks this is a N beat.

N?

**Correctness:** The heartbeat is actually a N heartbeat. Did the AI classify this heartbeat correctly?

◯ Yes

◯ No

## Explanation 1: Heatmap

Here is a heatmap of where in the heartbeat the AI thinks are most important.

## Explanation 2: Training Examples

The AI thinks this heatbeat is similar to other already known examples of N heartbeats.
They make look very similar but they are actually all different heartbeats.

# Task 7

**An AI algorithm thinks this is a F beat.**

**Correctness:** The heartbeat is actually a N heartbeat. Did the AI classify this heartbeat correctly?

○ Yes

○ No

## Explanation 1: Heatmap

Here is a heatmap of where in the heartbeat the AI thinks are most important.

## Explanation 2: Positive/Negative Analysis

The AI decision was influenced positively by the magenta areas and negatively by blue areas.

**Which explanation is better?**

◯ Heatmap (top)

◯ Positive/Negative Analysis (bottom)

---

# Task 8

## An AI algorithm thinks this is a N beat.

**Correctness:** The heartbeat is actually a N heartbeat. Did the AI classify this heartbeat correctly?

◯ Yes

◯ No

## Explanation 1: Hot Pixels

Magenta indicates the exact moments the AI thinks are important.

## Explanation 2: Training Examples

The AI thinks this heatbeat is similar to other already known examples of N heartbeats.
They make look very similar but they are actually all different heartbeats.

---

**Which explanation is better?**

◯ Hot Pixels (top)

◯ Training Examples (bottom)

---

# Task 9

## An AI algorithm thinks this is a N beat.

N?

**Correctness:** The heartbeat is actually a N heartbeat. Did the AI classify this heartbeat correctly?

⚪ Yes

⚪ No

# Explanation 1: Training Examples

The AI thinks this heatbeat is similar to other already known examples of N heartbeats.
They make look very similar but they are actually all different heartbeats.

# Explanation 2: Positive/Negative Analysis

The AI decision was influenced positively by the magenta areas and negatively by blue areas.

# Task 10

**An AI algorithm thinks this is a S beat.**

**Correctness:** The heartbeat is actually a N heartbeat. Did the AI classify this heartbeat correctly?

○ Yes

○ No

## Explanation 1: Hot Pixels

Magenta indicates the exact moments the AI thinks are important.

## Explanation 2: Positive/Negative Analysis

The AI decision was influenced positively by the magenta areas and negatively by blue areas.

> **Which explanation is better?**
>
> ○ Hot Pixels (top)
>
> ○ Positive/Negative Analysis (bottom)

---

# Task 11

## An AI algorithm thinks this is a N beat.

> **Correctness:** The heartbeat is actually a N heartbeat. Did the AI classify this heartbeat correctly?
>
> ○ Yes
>
> ○ No

## Explanation 1: Hot Pixels

Magenta indicates the exact moments the AI thinks are important.

## Explanation 2: Heatmap

Here is a heatmap of where in the heartbeat the AI thinks are most important.

**Which explanation is better?**

◯ Hot Pixels (top)

◯ Heatmap (bottom)

# Task 12

**An AI algorithm thinks this is a N beat.**

N?

---

**Correctness:** The heartbeat is actually a N heartbeat. Did the AI classify this heartbeat correctly?

○ Yes

○ No

---

## Explanation 1: Heatmap

Here is a heatmap of where in the heartbeat the AI thinks are most important.

## Explanation 2: Training Examples

The AI thinks this heatbeat is similar to other already known examples of N heartbeats.
They make look very similar but they are actually all different heartbeats.

**Which explanation is better?**

○ Heatmap (top)

○ Training Examples (bottom)

Note: only click submit once, or your entry may be lost.

Submit



[Supplementary Material 3 · image.pdf]



# Thanks for participating in our study!

You were selected as a possible participant in this study because you are an MTurk worker, and your participation in this research study is voluntary.

**Please read:**

For each task, you'll see an **image** and an **Artificial Intelligence (AI) prediction** of what is in the image.

First, we want to know if you think the AI was correct in its decision.

Next, there will be two methods explaining why the AI made this decision.

We want your opinion on **which is a better explanation** for this decision.

Please be sure to select one of the methods for every image.

Note: The images are a low resolution. Please do your best.

---

## Task 1

**An AI algorithm thinks this is a horse.**

**Correctness:** Do you think the AI classified this image correctly?

○ Yes

○ No

## Explanation 1: Hot Pixels

These are the pixels the AI thinks are important. Red indicates important areas.

## Explanation 2: Outline

The AI outlined the important parts of this image.

**Which explanation is better?**

◯ Hot Pixels (top)

◯ Outline (bottom)

---

# Task 2

**An AI algorithm thinks this is a dog.**

dog?

---

**Correctness:** Do you think the AI classified this image correctly?

○ Yes

○ No

---

# Explanation 1: Heatmap

Here is a heatmap of what the AI think is important. Red indicates important areas.

# Explanation 2: Masked Image

The visible parts are what the AI thinks are important

**Which explanation is better?**

◯ Heatmap (top)

◯ Masked Image (bottom)

# Task 3

**An AI algorithm thinks this is a truck.**

truck?

**Correctness:** Do you think the AI classified this image correctly?

◯ Yes

◯ No

## Explanation 1: Training Examples

The AI thinks this image is similar to other known images of trucks.

## Explanation 2: Outline

The AI outlined the important parts of this image.

**Which explanation is better?**

○ Training Examples (top)

○ Outline (bottom)

# Task 4

## An AI algorithm thinks this is a dog.

dog?

**Correctness:** Do you think the AI classified this image correctly?

○ Yes

○ No

## Explanation 1: Training Examples

The AI thinks this image is similar to other known images of dogs.

## Explanation 2: Heatmap

Here is a heatmap of what the AI think is important. Red indicates important areas.

**Which explanation is better?**

◯ Training Examples (top)

◯ Heatmap (bottom)

# Task 5

## An AI algorithm thinks this is a frog.

frog?

> **Correctness:** Do you think the AI classified this image correctly?
>
> ○ Yes
>
> ○ No

## Explanation 1: Heatmap

Here is a heatmap of what the AI think is important. Red indicates important areas.

## Explanation 2: Outline

The AI outlined the important parts of this image.

**Which explanation is better?**

◯ Heatmap (top)

◯ Outline (bottom)

# Task 6

**An AI algorithm thinks this is a truck.**

**Correctness:** Do you think the AI classified this image correctly?

◯ Yes

◯ No

## Explanation 1: Masked Image

The visible parts are what the AI thinks are important

## Explanation 2: Hot Pixels

These are the pixels the AI thinks are important. Red indicates important areas.

**Which explanation is better?**

○ Masked Image (top)

○ Hot Pixels (bottom)

---

# Task 7

## An AI algorithm thinks this is a airplane.

airplane?

**Correctness:** Do you think the AI classified this image correctly?

○ Yes

○ No

## Explanation 1: Positive/Negative Analysis

The AI decision was influenced positively by red pixels and negatively by blue pixels.

## Explanation 2: Masked Image

The visible parts are what the AI thinks are important

**Which explanation is better?**

◯ Positive/Negative Analysis (top)

◯ Masked Image (bottom)

---

# Task 8

**An AI algorithm thinks this is a bird.**

bird?

---

**Correctness:** Do you think the AI classified this image correctly?

◯ Yes

◯ No

---

## Explanation 1: Heatmap

Here is a heatmap of what the AI think is important. Red indicates important areas.

## Explanation 2: Positive/Negative Analysis

The AI decision was influenced positively by red pixels and negatively by blue pixels.

Which explanation is better?

◯ Heatmap (top)

◯ Positive/Negative Analysis (bottom)

# Task 9

**An AI algorithm thinks this is a horse.**

**Correctness:** Do you think the AI classified this image correctly?

◯ Yes

◯ No

## Explanation 1: Positive/Negative Analysis

The AI decision was influenced positively by red pixels and negatively by blue pixels.

## Explanation 2: Hot Pixels

These are the pixels the AI thinks are important. Red indicates important areas.

> **Which explanation is better?**
>
> ○ Positive/Negative Analysis (top)
>
> ○ Hot Pixels (bottom)

---

# Task 10

## An AI algorithm thinks this is a automobile.

> **Correctness:** Do you think the AI classified this image correctly?
>
> ○ Yes
>
> ○ No

## Explanation 1: Training Examples

The AI thinks this image is similar to other known images of automobiles.

## Explanation 2: Masked Image

The visible parts are what the AI thinks are important

**Which explanation is better?**

◯ Training Examples (top)

◯ Masked Image (bottom)

# Task 11

**An AI algorithm thinks this is a ship.**

ship?

---

**Correctness:** Do you think the AI classified this image correctly?

○ Yes

○ No

---

# Explanation 1: Outline

The AI outlined the important parts of this image.

# Explanation 2: Masked Image

The visible parts are what the AI thinks are important

**Which explanation is better?**

○ Outline (top)

○ Masked Image (bottom)

---

# Task 12

**An AI algorithm thinks this is a frog.**

**Correctness:** Do you think the AI classified this image correctly?

○ Yes

○ No

## Explanation 1: Outline

The AI outlined the important parts of this image.

## Explanation 2: Positive/Negative Analysis

The AI decision was influenced positively by red pixels and negatively by blue pixels.

**Which explanation is better?**

◯ Outline (top)

◯ Positive/Negative Analysis (bottom)

---

# Task 13

## An AI algorithm thinks this is a airplane.

**Correctness:** Do you think the AI classified this image correctly?

◯ Yes

◯ No

## Explanation 1: Hot Pixels

These are the pixels the AI thinks are important. Red indicates important areas.

## Explanation 2: Training Examples

The AI thinks this image is similar to other known images of airplanes.

airplane?          airplane          airplane          airplane

> **Which explanation is better?**
>
> ◯ Hot Pixels (top)
>
> ◯ Training Examples (bottom)

# Task 14

**An AI algorithm thinks this is a bird.**

bird?

**Correctness:** Do you think the AI classified this image correctly?

○ Yes

○ No

## Explanation 1: Training Examples

The AI thinks this image is similar to other known images of birds.

## Explanation 2: Positive/Negative Analysis

The AI decision was influenced positively by red pixels and negatively by blue pixels.

**Which explanation is better?**

◯ Training Examples (top)

◯ Positive/Negative Analysis (bottom)

---

# Task 15

**An AI algorithm thinks this is a deer.**

deer?

**Correctness:** Do you think the AI classified this image correctly?

◯ Yes

◯ No

## Explanation 1: Heatmap

Here is a heatmap of what the AI think is important. Red indicates important areas.

# Explanation 2: Hot Pixels

These are the pixels the AI thinks are important. Red indicates important areas.

**Which explanation is better?**

◯ Heatmap (top)

◯ Hot Pixels (bottom)

Note: only click submit once, or your entry may be lost.

Submit



[Supplementary Material 4]



# Thanks for participating in our study!

You were selected as a possible participant in this study because you are an MTurk worker, and your participation in this research study is voluntary.

**Please read:**

For each task, you'll watch an audio clip from someone speaking a word. An **Artificial Intelligence (AI) prediction** is trying to determine what word they are saying.

The person may say one of the following words: "yes", "no", "on", "off", "up", "down", "left", "right", "stop", "go".

First, we want to know if the AI was correct in its decision.

Next, there will be two methods explaining why the AI made this decision.

We want your opinion on **which is a better explanation** for this decision.

Please be sure to select one of the methods for every task.

NOTE: The videos may load slow, or may appear green. Please click them and they should play.

---

## Task 1

**An AI algorithm thinks this person is saying "right".**

**Correctness:** Did the AI classify this word correctly?

◯ Yes

◯ No

## Explanation 1: Heatmap

Here is a heatmap of where in the audio the AI thinks are most important.

## Explanation 2: Training Examples

The AI thinks this audio is similar to these three other known examples of someone saying 'right'.

> **Which explanation is better?**
>
> ◯ Heatmap (top)
>
> ◯ Training Examples (bottom)

---

# Task 2

**An AI algorithm thinks this person is saying "yes".**

**Correctness:** Did the AI classify this word correctly?

◯ Yes

◯ No

## Explanation 1: Heatmap

Here is a heatmap of where in the audio the AI thinks are most important.

## Explanation 2: Positive/Negative Analysis

The AI decision was influenced positively by the magenta areas and negatively by blue areas.

# Task 3

**An AI algorithm thinks this person is saying "down".**

---

| Correctness: Did the AI classify this word correctly? |
| --- |
| ◯ Yes |
| ◯ No |

## Explanation 1: Hot Pixels

Magenta indicates the exact moments the AI thinks are important.

## Explanation 2: Positive/Negative Analysis

The AI decision was influenced positively by the magenta areas and negatively by blue areas.

**Which explanation is better?**

○ Hot Pixels (top)

○ Positive/Negative Analysis (bottom)

# Task 4

# An AI algorithm thinks this person is saying "off".

<div>

**Correctness:** Did the AI classify this word correctly?

○ Yes

○ No

</div>

## Explanation 1: Hot Pixels

Magenta indicates the exact moments the AI thinks are important.

## Explanation 2: Training Examples

The AI thinks this audio is similar to these three other known examples of someone saying 'off'.

**Which explanation is better?**

◯ Hot Pixels (top)

◯ Training Examples (bottom)

# Task 5

**An AI algorithm thinks this person is saying "up".**

<br>

**Correctness:** Did the AI classify this word correctly?

○ Yes

○ No

# Explanation 1: Hot Pixels

Magenta indicates the exact moments the AI thinks are important.

# Explanation 2: Heatmap

Here is a heatmap of where in the audio the AI thinks are most important.

**Which explanation is better?**

○ Hot Pixels (top)

○ Heatmap (bottom)

---

# Task 6

**An AI algorithm thinks this person is saying "down".**

**Correctness:** Did the AI classify this word correctly?

◯ Yes

◯ No

# Explanation 1: Training Examples

The AI thinks this audio is similar to these three other known examples of someone saying 'down'.

# Explanation 2: Positive/Negative Analysis

The AI decision was influenced positively by the magenta areas and negatively by blue areas.

## Which explanation is better?

○ Training Examples (top)

○ Positive/Negative Analysis (bottom)

Note: only click submit once, or your entry may be lost.

Submit