[Reviews · NeurIPS 2020]

Review 1

Summary and Contributions: The paper asserts, from - an experimental evaluation based on approximately 500 participants, - audio, image, ECG and text data - a unified explanation framework for these data that explanation-by-examples is the most relevant scenario for the average non-technical end-users.

Strengths: Multimodal (but modality-independent) DNN explanation framework.

Weaknesses: The term ‘unified’ should be revised as the paper addresses a partial unification. For instance, the unified framework does not take into account a closed loop between the DNN and the explanation method (the explanation method can be itself another DNN interacting in a double sense with the prediction DNN) or other two-stage adaptive networks [1], [2]. In addition, an alternative to example based explanation is ‘opening the black box’ in terms of intra-layer and inter-layer statistical properties of a DNN [3]: these may be enough to explain lack of generality (and thus absence of recommendation) of a given network depending on the input available data and the classification paradigm considered. Thus, a positioning must be provided with respect to the above issues in order to make the paper more informative with respect to the literature. The weak spots of the analysis are twofold. On the one hand, the performance evaluation benchmark: no details are given with respect to the ages, activities and complementary of the participants so that the analysis can be severely biased (restricted to a specific public)! On the other hand, visualization, despite that it is an interesting option, is not necessary for explanation. Indeed, in a riddle, one may not be able to visualize the feature examples and be able to explain correct answers by cognition, causality and deductions. Therefore, fundamental research should necessarily be included in a formalism of unifying explainability frameworks, which is not currently the case in this paper. Minor comment: Domain notation D is useful: there is explicitly no run on D and no sub-domain information fusion so that omitting D has no impact on the equations. [1] Chen, Jie and Vaughan, Joel and Nair, Vijay and Sudjianto, Agus, Adaptive Explainable Neural Networks (Axnns), DOI: 10.2139/ssrn.3569318 [2] Joel Vaughan and Agus Sudjianto and Erind Brahimi and Jie Chen and Vijayan N. Nair, Explainable Neural Networks based on Additive Index Models, arXiv eprint 1806.01933, 2018. [3] A. M. Atto, R. Bisset and E. Trouve, Frames Learned by Prime Convolution Layers in a Deep Learning Framework, IEEE TNNLS 2020, DOI: 10.1109/TNNLS.2020.3009059

Correctness: There is no (questionable) theoretical contribution. Experimental claims concerning representativeness of experimental setup remain to be proved.

Clarity: Yes

Relation to Prior Work: Partially, see "weakness section"

Reproducibility: Yes

Additional Feedback:


Review 2

Summary and Contributions: The paper conducts non-expert human/end-user surveys to compare the explanations provided by different types of deep classifier explainers. These types include input feature attribution ones such as GradCAM++, black-box methods such as LIME and explanation-by-example approaches. The survey is conducted across four different domains of image (object classification), audio (keyword recognition), text (sentiment analysis), and signal processing (ECG arrhythmia diagnosis). One dataset was used for each of the tasks. All explanations of the different types are visually demonstrated to the end-users in a specific way chosen and argued for by the paper. The visual explanation was typically shown by the original input with the explanation superimposed on it. For the explanation-by-example type, the examples are directly shown. End-users were shown two types of explanations for some input examples and they had to choose which one they preferred. End-users were 455 non-expert Amazon M-Turk participants making a total of 4970 comparisons across all tasks and methods. The results indicate that explanation-by-example is preferred in all domains except text. For the text domain, LIME is preferred most often. Finally, some general discussions are made for and against the explanation methods and their usability for different scenarios.

Strengths: + The paper approaches the evaluation of the explanation methods purely from an end-user experience perspective. While this type of evaluation might raise concerns due to various subjectivities it is prone to, it can be quite complementary to the quantitative approaches we’ve had in the literature so far. + While in the hindsight (especially after looking at examples) it is intuitive that end-users prefer an explanation by examples if similar-enough examples are shown, it was surprising to fathom this observation the first time the quantitative results were provided in the paper’s introduction. + Interesting and important general discussions appear at the end of the paper including user privacy and availability of prototypical training data for explanation-by-example methods, importance of hyper-parameter optimization for LIME, Anchor, and explanation-by-example methods in contrast to the off-the-shelf usability of some others, stability of explanations, among others.

Weaknesses: - There are several design choices that could have had a game-changer effect on the human study. This primarily includes the choice of the way the explanation results are illustrated for various methods on different datasets/tasks. Questions like the following can be asked: would Shap receive a better rating if the superimposition was of higher resolution? Would attribution techniques receive a better rating if the explanations were demonstrated using different/less colors? A suggestion here could be to design and use various options in the human study and include the research question of “what is the best visualization of an explanation type” as part of the human study since it seems to be hard to disentangle the two. - The reviewer is not sure about the pertinence of the use of non-expert end-users in the human study, especially for expert decisions such as arrhythmia diagnosis using ECGs. Wouldn’t it be better to conduct such studies with the users that are actually going to be interested and are able to judge explanations appropriately? - The number of datasets and number of comparisons seem too limited for a human study. The reviewer expected orders of magnitude more comparisons (Than ~5000) especially when using non-expert M-Turk users.

Correctness: Yes.

Clarity: The paper is clearly written and it was easy to read through it.

Relation to Prior Work: ample

Reproducibility: Yes

Additional Feedback: --- after rebuttal - I read the rebuttal, and discussed among the reviewers. I keep my original borderline rating since I think explanation techniques have important real-world applications, and while there are several works that introduce new explanation techniques, there is no consensus on the optimal way the explanation methods should be evaluated, so this work is a step in a complementary direction that can help shape the future works by conducting evaluations through user study. On the other hand, I do see that the work could be improved in the way the study is done or the results are interpreted. So, I remain at borderline accept.


Review 3

Summary and Contributions: This paper presents results from a user study on existing explanation methods for machine learning models, in order to determine which type of explanation is best suited for a particular application domain. The authors conducted an AMT study across six popular XAI methods and four distinct application domains, and qualitatively assess which were preferred by different target users.

Strengths: The problem of explaining neural network models in a human-intuitive manner is certainly important, relevant to NeurIPS, and subject of ongoing research. The paper presents some interesting initial findings about user preferences for different types of explanations.

Weaknesses: The problem the authors are trying to solve, of ranking explanation methods based on user preference, is interesting and important but not well-founded. While this effort is a good first step, there are many considerations to be made in designing user studies to assess the interpretability of explanations for machine learning models. My main concern is that for data from domains that requires some understanding (so other than natural images, speech, non-technical text, etc.), asking an average user which explanation they prefer is an ill-posed question. My second concern is that using explanation methods originally designed for image classification models to explain time series data (waveform for audio, EEG) may not be suited to produce a human-interpretable explanation and that would certainly affect what the users prefer. My last concern is about the novelty and scope of this work: for the image domain, datasets like CIFAR100 and ImageNet should be included in the study, and other explanation methods (I mention some below) also added. More concretely on the user study remark: I think the problem of assessing end-user preferences for explanations produced by different types of explanation methods is very delicate. Without significant domain knowledge, there will be discrepancy between user preferences due to human bias in perception and interpretation. That does not seem to be accounted for: users are discarded if they respond to survey questions too quickly, and are also asked whether the user thinks the underlying model classified the input correctly after providing the user with the prediction. However, asking a user to rank two types of explanations based on their preference is a task susceptible to a high degree of noise, without taking into account also how often the user agreed that the model predicted the output correctly. To clarify my point, I refer the authors to a recent paper on explanations for image similarity models that designs a user study in a slightly different manner; asking a user to guess whether the model predicts the correct classification based purely on different types of explanations, and having a control case of no explanations provided as a baseline; then, measuring relative accuracy of user answers across different explanations would provide more information about their helpfulness. See Plummer et al., “Why do These Match? Explaining the Behavior of Image Similarity Models”, for a study along those lines. I am also not so sure about casting EEG and audio data such that explanation systems originally designed to explain image classification models could be applied: does that make sense? To the average user, these signals in image form (eg., waveform) would be significantly less intelligible than a natural image. As such, there is no surprise that example-by-explanation is highly preferred, since users may struggle to understand what is supposed to be important for the model’s prediction by simply looking at a waveform. One more question I wondered about is, in explanation-by-example methods, is the NN query approach standard? As you mention in the paper, the explanation is sometimes counterintuitive (eg., predict “airplane” and highlight the sky as the salient region) - does that consistently hold true across the retrieved examples for explanation? Have you checked this? (as in, are those also going to be images where the sky is highlighted as most important). Analogically for other modalities. Basically, if I show the user an image of an airplane in the sky and the sky is the most salient region according to, say, saliency maps or LIME, when I am comparing those methods against explanation-by-example approaches and the NN matching is done simply based on the feature representations from the last conv layer (line 153), that seems to give the method unfair advantage and it’s natural for a user to pick that explanation as better. Lastly, the choice of explanation methods evaluated from each category could be better motivated: eg., did you select certain methods because of popularity? Ease of implementation? Particular application in mind?

Correctness: For the EEG setup, the questions in the user study provide the actual answer to the user and ask if the model was correct in its prediction. However, under that setup, the study measures more so whether the model was accurate rather than whether the explanation was sufficiently informative/helpful to the user in understanding what the model predicts. When the authors say “we empirically observed the last convolutional layer to consistently provide the most well-formed explanations” for the model-transparent approaches like GradCAM++, saliency maps, SHAP, and explanation by example, what do you mean by “well-formed”? Are you referring to the degree of alignment between the localization of salient regions with your intuition? Can this be quantified? Also since as you point out in later paragraphs, human intuition for what constitutes a good explanation does not always coincide with what the most useful information is for the model to make its prediction, I would be wary of making qualitative assessment. On that note, Section 4.2 discusses exactly this problem of human intuition of what is important, vs. important parts of the image to the model. If your study simply asks users whether they like a particular explanation, the explanation method’s assessment of what is important in an image is being scored against the human intuition of what is important, and you would be answering the question “which explanation approach provides explanations humans agree with the most, on average”. I don’t believe this question assesses the actual quality of the explanations produced by these methods across data/tasks of different domains. For the image domain portion, comparing methods that produce more localized, compact explanations like Fong and Vedaldi, “Interpretable Explanations of Black Boxes by Meaningful Perturbation”, or Petsiuk et al., “RISE: Randomized Input Sampling for Explanation of Black-box Models”, would be more informative in the user study.

Clarity: The paper is well written but suffers from technical issues. (minor: line 159: “supported” repetition)

Relation to Prior Work: The authors provide a somewhat comprehensive overview of the different types of explanation methods but could certainly extend that list in their study.

Reproducibility: Yes

Additional Feedback: -- post-rebuttal feedback -- Thank you for your response. After careful review, the concerns I have with evaluation methodology, choice of fair baselines, and scale and formulation of the user study remain as they can only be answered by redesign. Without that, I find the practical usefulness of the conclusions in the paper limited, and the takeaways possibly misleading. Another route to improvement is to expand on sections 4.1 and 4.2, where a lot of hypothetical interpretation of the results are offered, and verify some of them with further studies. That is currently a gap and would be a very useful contribution to the XAI literature. Overall, I decided on keeping my original score and hope that the importance of carefully thinking around evaluation methodology when involving human subjects is acknowledged, as it sets the entire premise of such papers and dictates the main takeaway.


Review 4

Summary and Contributions: This paper performs an empirical study of multiple explanation methods to test which ones are "more preferable" to an end-user (AMT workers). The authors test multiple explanation methods (gradient based, prototype based and black box) on multiple domains (image, text, audio and ECG signals). They find that users in general prefer a prototype based method over other types. Authors release the user responses and the code.

Strengths: 1. The authors study multiple input domains across explanation methods from various class types. 2. Human evaluation of explanations (and their preference in general) is still not as tested (compared to explosion of interpretability methods). Therefore, it provides a significant data point towards moving away from standard saliency based methods to other domains -- prototypes being the most important example.

Weaknesses: 1. Mathiness: The authors try to ground there unification of explanation methods in some mathematical notation that took quite a bit of time to parse. More importantly, it is neither correct not required and only takes away from the contributions. For example authors say inputs lie in real domain -- not true for text. Do the elements lie in R^p or the whole dataset itself (each element lie in R^(p/|D|) ?). Why is visual domain R^v (shouldn't it be 2-D?) -- and what is visual domain anyway ? I would strongly suggest removing this math and just explain it in words . 2. The study asks which explanation is better -- is better defined somewhere ? Was the goal to be so open ended -- in which, one would need to know what the end user thinks the explanation is telling them ? There are other studies that also test the human preference but along concrete dimensions (eg. https://arxiv.org/pdf/1901.03729.pdf, https://www.aclweb.org/anthology/D19-1523/). In general, while it is an interesting data point, it is not particularly useful to making decisions in any real application. (NOTE to AC -- subsequent papers appeared after neurips deadline so not missing citations) I would also suggest looking at https://arxiv.org/abs/2005.01831, https://arxiv.org/abs/2005.00115, https://arxiv.org/pdf/2006.15631.pdf for how to make the study questions a bit more specific. 3. Important citations missing -- see previous point for example. In addition, more prototype based methods (https://arxiv.org/abs/1806.10574, https://arxiv.org/abs/1906.10651).

Correctness: See previous. I still find the claims to be overbroad due to very open-ended study design. The methods within the parameters of the current study is okay. I don't find any glaring mistakes in statistical methodology. I would have added a learning period before asking users to perform this study so they are familiar with the task and the explanation methods outputs (not the method themselved).

Clarity: Yes, except for mathiness issues.

Relation to Prior Work: See weaknesses. In general, work don't repeat any existing studies in this domain.

Reproducibility: Yes

Additional Feedback: Please also add if you used IRB approval for this project (would most likely be exempt, but since you are asking for user's subjective opinions, It is some cases needed). Also please provide more details about how workers were selected, how much they were paid in main paper.

[Author Response · NeurIPS 2020]

Thank you all for your thoughtful reviews. The following document seeks to address some reviewer concerns:

**Motivation and Contribution (All):** Reviewer 3 succinctly summarized our contribution in his review: "which explanation approach provides explanations humans agree with the most, on average." Our study was not intended to reveal which explanation is of superior quality; we apologize if that was the perceived motivation. Instead, the aim is to offer insight as to which of the popular explanation methods are more appealing to an average end-user who may not possess knowledge of machine learning, i.e. the "non-expert" layperson. This distinction will be made more explicit in a camera-ready draft.

**Visual Explanations (Rev 1, 3):** Reviewers 1 and 3 astutely note that explanations are not inherently constrained to the visual domain. We acknowledge and agree with these observations; while not explicitly stated, our study was primarily restricted to the visual domain. Given that visual explanations are the most widely-used of available explanation methods, we restricted our focus to these methods, thereby limiting the overall scope of the study. When referring to "visual explanations" in our analysis and unification, we refer to an explanation that is perceived through sight (including text), not that the explanation is necessarily an image. Future works that introduce explanation methods spanning alternative domains would certainly mandate further analysis. We would be sure to clarify this constraint in a camera-ready submission.

**Casting Other Domains to Image Classification Explanations (Rev 3):** As Reviewer 3 mentions, casting image explanations to time-series data is undoubtedly a sub-optimal solution. Nevertheless, a survey of the current literature revealed that these explanation methods are frequently applied to these domains [1][2][3]. An emphasis of the shortcomings of existing solutions coupled with the need for explanation methods specifically targeting these input domains will be included in a camera-ready submission.

**Unification of Explanation Presentation (Rev 1, 4):** We agree with Reviewer 1 that our unification does not encapsulate explanations from interpretable architectures, adaptive NNs, and methods exploring the intra-layer and inter-layer statistical properties. Our unification encompasses visual explanation methods for open-loop, forward-path inferences given a pre-trained model and a test input. We will be sure to offer an appropriate positioning of our unification with respect to the literature and focus more on improving the textual explanation, as per the advice of Reviewer 4.

**IRB Approval and Study Size (Rev 2, 4):** Our study was submitted for IRB approval and received exemption. We chose to compensate participants according to the advocated minimum wage of $15 per hour [4]. Given our limited budget, this restricted the scale of our study, but nevertheless provided statistically significant results when considering either bootstrap confidence or population sampling confidence intervals. Related works surveyed at a similar magnitude [5]. These details will be included in our camera-ready submission.

**Ensuring a Fair Comparison (Rev 3):** For all of the explanation methods that required a specific layer input to produce an explanation, we universally selected the last convolutional layer to ensure a fair comparison. This selection for saliency methods is grounded in recommendations from prior work (please refer to submission citation 8). As such, the same layer is selected for the NN-query in explanation-by-example. Our survey validation questions specifically accounted for how often the user agreed with the model prediction; we only removed fast submissions if they were physically impossible to achieve (e.g. the survey participant used a bot to autocomplete the form). The statement regarding "well-formed" was not a quantitative assessment and will be retracted.

**Methods Selected (Rev 3):** We agree with Reviewer 3 that including localized and compact explanations for image classification are desirable for a more informative study. It is for this reason that we specifically selected Grad-CAM++ over Grad-CAM, as it also advocates for a more localized and compact explanation. As opposed to evaluating across saliency methods specifically, our study was intended to provide a comparative analysis across the various approaches to explanation as they are perceived by the average human. Given the wide body of available methods in the literature, we aimed to select the necessary but sufficient subset of methods that were most commonly explored in related works and explainability studies. The NN-query method specifically used in the explanation-by-example implementation is based on a number of prior works (e.g. [1], submission citation 17).

**Related Works (All):** Thank you all for pointing out these important related works. We will expand on our related works section to include all references to explainable NNs, methods that observe statistical properties of a DNN, other saliency methods, and the related explainability studies. We will be sure to include them in the camera-ready submission.

[1] Gee et al. "Explaining deep classification of time-series data with learned prototypes" 2019
[2] Arnout et al. "Towards A Rigorous Evaluation Of XAI Methods On Time Series" ICCVW 2019
[3] Assaf et al. "Explainable Deep Neural Networks for Multivariate Time Series Predictions" IJCAI 2019
[4] Williamson et al. "On the Ethics of Crowdsourced Research" PS: Political Science & Politics, vol. 49, no. 1, 2016
[5] Hase et al. "Evaluating Explainable AI: Which Algorithmic Explanations Help Users Predict Model Behavior" 2020


[Meta-Review · NeurIPS 2020]

The paper is an empirical study on the types of explanations preferred by users using AMT. All reviewers found that problem was important and found the study interesting. However, one reviewer argued that while this was a good first step, it does not address the fact evaluating explanations is an ill-posed problem. Three reviewers found that this study is interesting enough to the NeurISP community even as a first step. Therefore, I recommend acceptance.